# Salt-inducible kinase 3 regulates the mammalian circadian clock by destabilizing PER2 protein

Naoto Hayasaka[1,2,3]*, Arisa Hirano[4], Yuka Miyoshi[3], Isao T Tokuda[5], Hikari Yoshitane[4], Junichiro Matsuda[6], Yoshitaka Fukada[4]

[1]Department of Neuroscience II, Research Institute of Environmental Medicine, Nagoya University, Nagoya, Japan; [2]PRESTO, Japan Science and Technology Agency, Kawaguchi, Japan; [3]Department of Anatomy and Neurobiology, Kindai University Faculty of Medicine, Osaka, Japan; [4]Department of Biological Sciences, School of Science, The University of Tokyo, Tokyo, Japan; [5]Department of Mechanical Engineering, Ritsumeikan University, Kusatsu, Japan; [6]Laboratory of Animal Models for Human Diseases, National Institutes of Biomedical Innovation, Health and Nutrition, Ibaraki, Japan

**Abstract** Salt-inducible kinase 3 (SIK3) plays a crucial role in various aspects of metabolism. In the course of investigating metabolic defects in *Sik3*-deficient mice (*Sik3*$^{-/-}$), we observed that circadian rhythmicity of the metabolisms was phase-delayed. *Sik3*$^{-/-}$ mice also exhibited other circadian abnormalities, including lengthening of the period, impaired entrainment to the light-dark cycle, phase variation in locomotor activities, and aberrant physiological rhythms. Ex vivo suprachiasmatic nucleus slices from *Sik3*$^{-/-}$ mice exhibited destabilized and desynchronized molecular rhythms among individual neurons. In cultured cells, *Sik3*-knockdown resulted in abnormal bioluminescence rhythms. Expression levels of PER2, a clock protein, were elevated in *Sik3*-knockdown cells but down-regulated in *Sik3*-overexpressing cells, which could be attributed to a phosphorylation-dependent decrease in PER2 protein stability. This was further confirmed by PER2 accumulation in the *Sik3*$^{-/-}$ fibroblasts and liver. Collectively, SIK3 plays key roles in circadian rhythms by facilitating phosphorylation-dependent PER2 destabilization, either directly or indirectly.

*For correspondence:
naotohayasaka@yahoo.co.jp

Competing interests: The authors declare that no competing interests exist.

## Introduction

Circadian rhythms in mammals are governed by the master oscillator located in the hypothalamic suprachiasmatic nucleus (SCN). A growing body of evidence suggests that oscillation of the circadian clock is generated by the transcriptional-translational feedback loop composed of the circadian clock genes and their products, including the transcriptional activators CLOCK and brain/muscle ARNT-like protein 1 (BMAL1), and their negative regulators PERIODs (PER1/2/3) and CHRYPTOCHROMEs (CRY1/2)(*Brown et al., 2012*; *Takahashi, 2015*). These factors and their modulators contribute to determination and fine-tuning of the circadian period and the phase. For example, *Cry1* and *Cry2* knockout (KO) mice, respectively, demonstrate shortened and lengthened periods of the locomotor activity rhythms in constant dark condition (*van der Horst et al., 1999*). *Tau* mutant hamster or mice harboring a missense mutation in casein kinase I epsilon (*Csnk1e, CKIε*) exhibit shortened circadian periods, and the phase was advanced in behavioral and physiological rhythms (*Meng et al., 2008*; *Ralph and Menaker, 1988*). These lines of evidence suggest that the transcriptional-translational feedback loop mediated by the clock genes, and the post-translational modification of their products, are indispensable to the circadian clock machinery. However, the molecular mechanisms

underlying the determination or stabilization of the circadian period and phase remain to be investigated in mammals.

Previous reports have suggested that protein kinases play important roles in the regulation of circadian clocks (*Reischl and Kramer, 2011*). For example, CSNK1d/e (CKIδ/ε) phosphorylates PER1 and PER2 proteins and promotes their proteasomal degradation (*Akashi et al., 2002*; *Eide et al., 2005*; *Meng et al., 2008*). CRY1 are also phosphorylated by 5′ AMP-activated protein kinase (AMPK) and DNA-dependent protein kinase (DNA-PK) (*Gao et al., 2013*; *Lamia et al., 2009*). CRY2 is specifically phosphorylated by dual-specificity tyrosine-phosphorylation-regulated kinase 1A (DYRK1A) and glycogen synthase kinase 3 beta (GSK3ß), leading to degradation via a ubiquitin-proteasomal pathway (*Gao et al., 2013*; *Kurabayashi et al., 2010*; *Lamia et al., 2009*). We previously reported that mitogen-activated protein kinase (MAPK/ERK) contributes to robust circadian oscillation within individual SCN neurons (*Akashi et al., 2008*). Furthermore, the significance of protein kinases in human circadian rhythms has also been studied. Toh *et al.* reported that a point mutation in the clock gene *Per2* results in familial advanced sleep phase syndrome (FASPS) in a human kindred (*Toh et al., 2001*). A patient with FASPS exhibits an inherited abnormal sleep pattern in which the circadian clock is phase-advanced by approximately 4 hr. The FASPS mutation causes hypo-phosphorylation of PER2 protein by CSNK1e. CSNK1e requires a priming kinase for phosphorylation, which is yet to be identified. Overall, these data suggest that post-translational modifications of clock proteins by a series of protein kinases play critical roles in regulation of the circadian clock. In the current study, we focused on salt-inducible kinase 3 (SIK3) because we found an abnormal circadian rhythmicity in metabolism while we were studying *Sik3* knockout (KO) mice (*Uebi et al., 2012*). SIK1-3 are Ser/Thr kinase members of the AMPK family, which is known as an energy sensor, and regulates various aspects of metabolism in both vertebrates and invertebrates (*Katoh et al., 2004*; *Wang et al., 2011*). Previous studies suggest critical roles for mammalian SIK3 in the metabolism of glucose, cholesterol, and bile acid, and in retinoid metabolism and skeletal development (*Sasagawa et al., 2012*; *Uebi et al., 2012*). *Sik3*−/− mice exhibit severe metabolic symptoms, such as hypolipidemia and hypoglycemia, and very frequently die on the first day after birth (*Uebi et al., 2012*). Furthermore, they exhibit severe defects in chondrocyte hypertrophy, expanded growth plate or cartilage, accumulation of chondrocytes, and impaired skull bone formation (*Sasagawa et al., 2012*). In the present study, we found that *Sik3* KO mice also demonstrate abnormal circadian phenotypes, which are similar in part, but distinct from those of *Csnk1* mutants.

## Results

### *Sik3*−/− mice exhibit abnormal circadian rhythms in physiology and behavior

While studying metabolic phenotypes in *Sik3*−/− mice, we observed that the average oxygen consumption rhythm, representing the circadian rhythmicity of metabolism, was significantly phase-delayed by approximately 6 hr compared with wild-type (WT) littermates (*Figure 1a*). We also found that other physiological rhythms related to rhythms related to food consumption were phase-delayed to similar degrees (*Figure 1b*).

These data raised the possibility that SIK3 plays a role in the circadian clock machinery. We performed in situ hybridization on mouse brain sections and found that *Sik3* mRNA is expressed in the SCN, the central circadian oscillator (*Figure 1—figure supplement 1a*, arrowhead), although *Sik3* mRNA levels are not regulated in a circadian manner (*Figure 2—figure supplement 1b*). We then performed behavioral analysis of the *Sik3*−/− mice and WT (*Sik3*+/+) littermates. In 12 hr light-dark (LD) cycle conditions (LD 12:12), *Sik3*+/+ mice exhibited nocturnal activity profiles in which they were entrained to the LD cycle, with activity onset at light-off (ZT12, zeitgeber time representing light-on as ZT0 and light-off as ZT12 in LD 12:12 conditions) and the activity offset at light-on (ZT0). In stark contrast, *Sik3*−/− mice failed to become completely entrained to the LD cycle (i.e. re-entrainment from DD to LD causes 'splitting' phenotype, in which a part of the locomotor activity is entrained to the LD cycle but the other part appears to be free running), although they did still retain nocturnality (*Figure 1d*, upper panels). Their activity onset and offset in the LD cycle did not correspond exactly with the light-off and light-on times, respectively, and they fluctuated significantly with regard to light-on/light-off. These data suggest that *Sik3*−/− mice have a defect in a light-input pathway. In

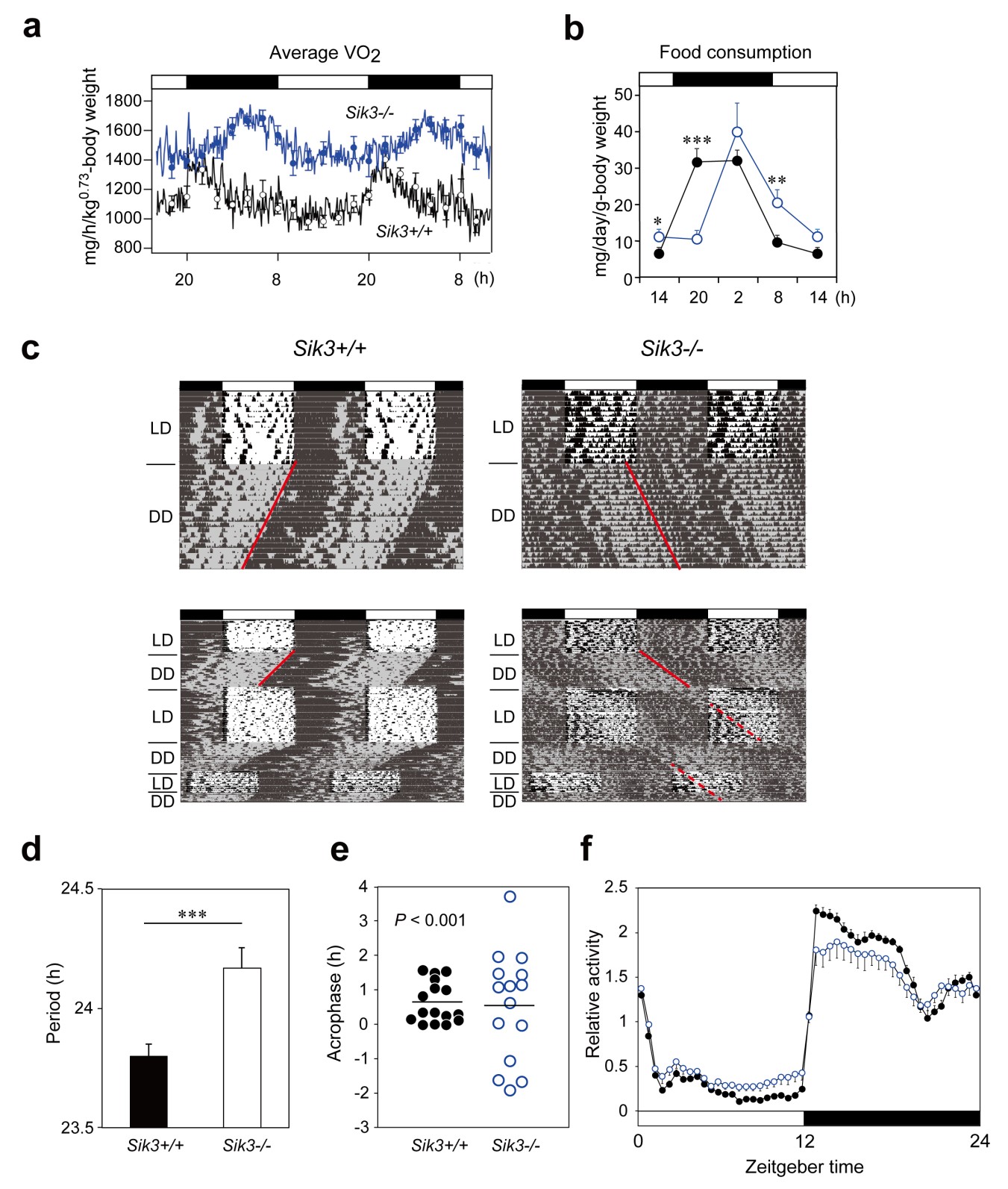

**Figure 1.** *Sik3⁻/⁻* mice exhibit aberrant circadian rhythms in physiology and behavior. (a) Oxygen consumption rhythm was measured in *Sik3⁻/⁻* mice and *Sik3⁺/⁺* controls for 2 days (*n* = 3 per group), and mean values are shown. The peak in the *Sik3⁻/⁻* mice (blue line) was phase-delayed by approximately 6 hr compared with that of *Sik3⁺/⁺* mice (black line). (b) Food consumption rhythm also exhibited delayed phase (*n* = 3 per each). Note that continuous food intake was observed in the *Sik3⁻/⁻* mice during the resting phase (daytime). (c) Locomotor activity rhythms in the *Sik3⁻/⁻* mice and *Sik3⁺/⁺* controls.

*Figure 1 continued on next page*

Figure 1 continued

Upper panels demonstrate impaired entrainment to the light-dark (LD) cycles, low amplitude, variable phases, and significantly longer free-running periods in $Sik3^{-/-}$ mice compared with $Sik3^{+/+}$ controls (see angles of red lines). Lower panels show light re-entrainment experiments. In contrast to $Sik3^{+/+}$ mice, in which behavioral rhythms entrained to LD cycles after transition from constant dark (DD) to LD, it took more than 2 weeks for $Sik3^{-/-}$ mice to become entrained to LD. In addition, a portion of the activity rhythms in DD persistently free-ran for as long as 3 weeks after conversion from DD to LD in the $Sik3^{-/-}$ mice (see dotted lines). (d) Average free-running periods in the $Sik3^{-/-}$ mice (n = 13) and $Sik3^{+/+}$ mice (n = 15). (e) Distribution of average acrophase in individual $Sik3^{-/-}$ (n = 15) and $Sik3^{+/+}$ (n = 16) mice. p<0.001 for the test for equality of variance (F-test) vs. $Sik3^{+/+}$. (f) Averaged activity plots of $Sik3^{+/+}$ mice (black line, n = 16, average of 16 days) and $Sik3^{-/-}$ mice (blue line, n = 15) in LD. *p<0.05, **p<0.01, ***p<0.001 vs. $Sik3^{+/+}$ (Student's t-test).

The online version of this article includes the following figure supplement(s) for figure 1:

**Figure supplement 1.** Expression patterns of the $Sik3$ mRNA in the mouse brain.

constant dark (DD) conditions, the free-running period of the wheel-running activity rhythms of the $Sik3^{-/-}$ mice (24.17 ± 0.09 hr [n = 13]) was significantly longer than that of $Sik3^{+/+}$ mice (23.80 ± 0.05 hr [n = 15]) (**Figure 1d,e**). The ability of $Sik3^{-/-}$ mice to entrain to a new LD cycle was examined by photic re-entrainment experiments, in which mice housed in DD conditions were transferred to an LD cycle, the phase of which was shifted substantially from the free-running activity rhythm. The free-running locomotor rhythms of $Sik3^{+/+}$ mice rapidly re-entrained to the new LD cycle, in contrast, $Sik3^{-/-}$ mice exhibited slower re-entrainment (**Figure 1d**, lower panels). Even after 2 weeks in the new LD cycle, a component of the free-running rhythm formed under the DD condition (red dotted line in **Figure 1d**, lower right panel) was still observed, although the majority of the locomotor rhythm components were entrained to the LD condition. The acrophase of the activities of individual $Sik3^{-/-}$ and $Sik3^{+/+}$ mice were then compared in LD (**Figure 1f**). Acrophase means did not differ significantly between the two genotypes, but the individual variation (standard deviation) in the acrophase distribution in $Sik3^{-/-}$ mice was significantly greater than that of $Sik3^{+/+}$ mice (p<0.001, F-test). These data indicated that SIK3 plays a crucial role in controlling and sustaining robust circadian behavioral rhythmicity in addition to contributing to the photic entrainment. Notably, $Sik3^{-/-}$ mice were relatively less active at night than $Sik3^{+/+}$ mice and more active during the day, which resulted in lower amplitude in the locomotor activity rhythms (**Figure 1d,g**). It should also be noted that significant differences in locomotor activity rhythms of individual $Sik3^{-/-}$ mice, which include circadian period, acrophase, and activity patterns at light-re-entrainment, were observed.

## Molecular rhythms are desynchronized and circadian periods and phases are varied in ex vivo SCN

To explore how SIK3 deficiency affects the circadian clocks in the SCN and periphery, $Sik3^{-/-}$ mice were crossed with $Per2::Luciferase$ knockin (KI) mice ($Per2^{Luc}$) expressing PER2-luciferase fusion protein in place of PER2, which monitors the circadian oscillation of PER2-luciferase fusion protein both in SCN explants and in cultured cells (**Yoo et al., 2004**). Brain slices including the SCN were prepared from $Sik3^{-/-}$; $Per2^{Luc/Luc}$ double-homozygous mice for ex vivo bioluminescence imaging at the single-cell level (**Figure 2**). As previously reported (**Akashi et al., 2008**; **Fukuda et al., 2011**; **Yoo et al., 2004**), $Sik3^{+/+}$ SCN explants exhibited robust circadian bioluminescence rhythms and the individual cells were well synchronized with one another (**Figure 2a,d**, **Figure 2—video 1**). In contrast, the circadian periods and phases of the individual cells in $Sik3^{-/-}$ SCN slices were varied and their bioluminescence rhythms became gradually desynchronized with time (**Figure 2e,h**, **Figure 2—video 2**). To further evaluate the difference in the SCN rhythms between the two genotypes, we performed quantitative analyses on the periods and phases of the imaging data (**Figure 2a–h**). The number of oscillating cells in $Sik3^{-/-}$ SCN, calculated as the percentage of oscillating pixels within the SCN, was significantly lower than that of $Sik3^{+/+}$ controls (95.9% in $Sik3^{+/+}$ SCN vs. 83.3% in $Sik3^{-/-}$ SCN, p<0.01 [chi-squared periodogram]). Although the average periods of the individual SCN cells in the $Sik3^{-/-}$ and $Sik3^{+/+}$ slices were comparable with one another, the variation (standard deviation) was significantly greater in the $Sik3^{-/-}$ cells than in the $Sik3^{+/+}$ cells (**Figure 2b,f,i**; p<0.0001 [F-test]). Similarly, the average acrophases of the $Per2^{Luc}$ rhythms of individual SCN cells were comparable between the two genotypes, but the variation in the $Sik3^{-/-}$ cells was significantly greater than that in the $Sik3^{+/+}$ cells (**Figure 2c,g,j**; p<0.0001 [F-test]). Overall, our data demonstrated that $Sik3$ deficiency reduced the number of oscillating cells and impaired the circadian rhythmicity of oscillating

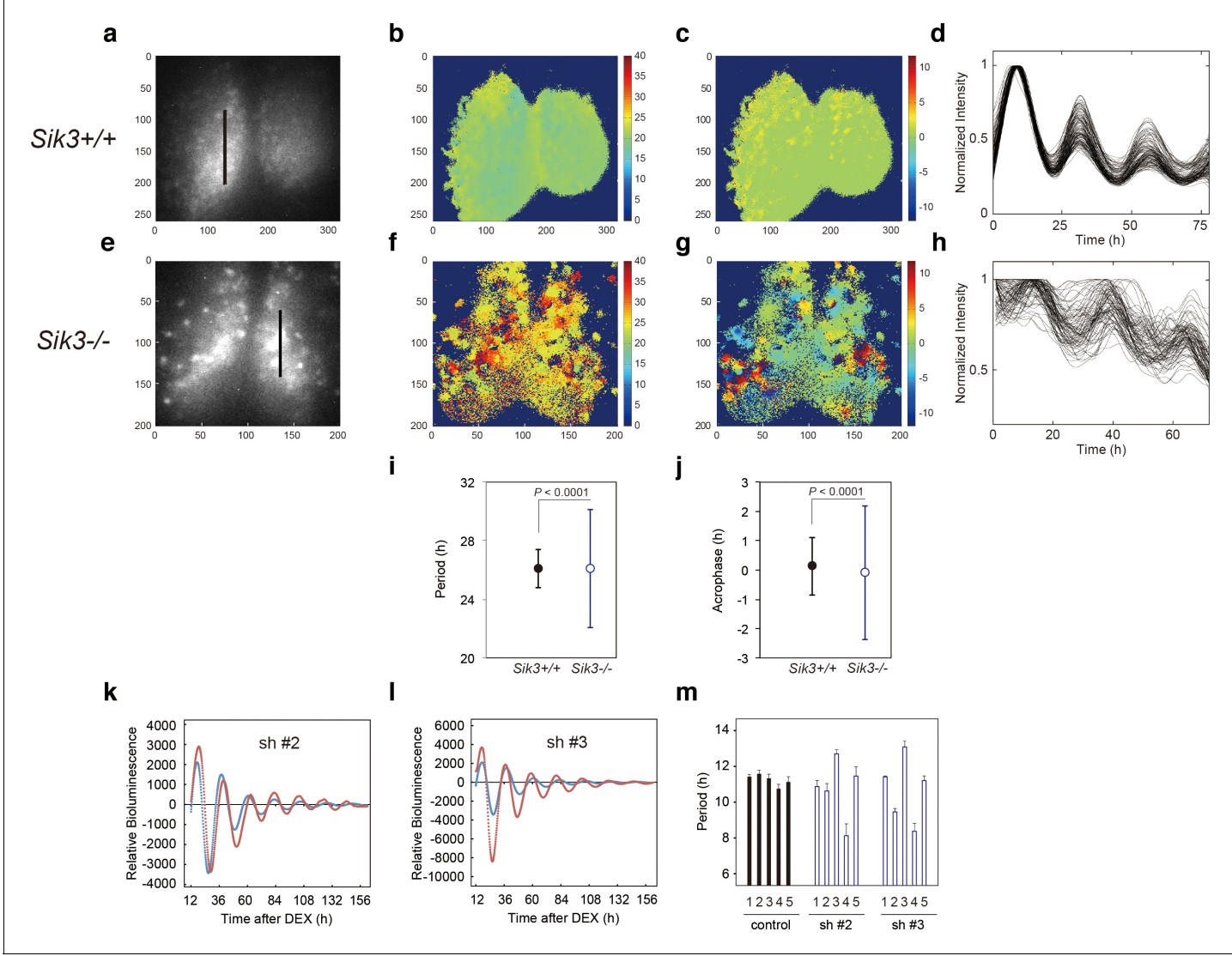

**Figure 2.** Varied circadian periods and phases in individual cells of the *Sik3*[-/-] suprachiasmatic nucleus (SCN). Time-lapse single-cell bioluminescence images of ex vivo SCN explant cultures of (a) *Sik3*[+/+]; *Per2*[Luc/Luc] and (e) *Sik3*[-/-]; *Per2*[Luc/Luc] mice were used for the following analyses: Period distribution on SCN slices of *Sik3*[+/+] (b) and *Sik3*[-/-] (f). Acrophase distribution on SCN slices of *Sik3*[+/+] (c) and *Sik3*[-/-] (g). Time series of the bioluminescence signals observed on a *Sik3*[+/+] SCN slice along the bold line indicated in a) (d), and those from a *Sik3*[+/+] SCN slice along the bold line indicated in (b) (h). The original signals were smoothed via a moving average filter, and then normalized. (i) Period distribution in representative *Sik3*[+/+] and control slices. p<0.0001 vs. *Sik3*[-/-] (test for equality of variance, F-test). (j) Acrophase distribution of representative *Sik3*[+/+] and *Sik3*[-/-] slices. p<0.0001 vs. *Sik3*[-/-] (F-test). (k) and l) Representative *Bmal1-luc* rhythms of *Sik3*-KD NIH3T3 (sh #2 and #3, red) and control cells (blue). (m) Averaged sequential half-periods (peak-to-trough or trough-to-peak) of *Bmal1-luc* rhythms in KD and control NIH3T3 cells as shown in (k) and l) (n = 4 per group). Numbers (1–5) on the x-axis present the first half-period (hours from first trough to second peak) to the fifth half-period (hours from third trough to fourth peak).

The online version of this article includes the following video and figure supplement(s) for figure 2:

**Figure supplement 1.** *Sik3* knockdown cells exhibit shortened and unstable period of circadian *Per2*[Lucc] rhythms.

**Figure 2—video 1.** Time-lapse bioluminescence imaging of a coronal SCN slice of a *Sik3*[+/+]; *Per2*[Luc/Luc] mouse.
https://elifesciences.org/articles/24779#fig2video1

**Figure 2—video 2.** Time-lapse bioluminescence imaging of a SCN slice of a *Sik3*[-/-]; *Per2*[Luc/Luc] mouse.
https://elifesciences.org/articles/24779#fig2video2

cells, that is, it was associated with instability in circadian period and phase, and desynchrony among SCN cells.

## In vitro Sik3-knockdown destabilizes circadian periods

Fluctuation of the circadian period and phase in $Sik3^{-/-}$ mice in vivo and ex vivo prompted the examination of whether alteration of SIK3 expression levels affected cellular circadian rhythms in culture. NIH3T3 cells were transfected with vectors expressing four different $Sik3$ short-hairpin RNAs (shRNAs), and cellular rhythms were monitored via a $Bmal1$-luc reporter gene (*Figure 2k,l*, *Figure 2—figure supplement 1a*). Unexpectedly, the average periods of all the $Sik3$-knockdown ($Sik3$-KD) cells were significantly shorter than those of WT controls (*Figure 2—figure supplement 1b*). Detailed period analyses revealed that all $Sik3$-KD cells exhibited asymmetry in the rhythmic pattern of $Bmal1$-luc half-periods, representing individual peak-to-trough or trough-to-peak periods, and fluctuated in all knockdown (KD) cells examined during 1 week of monitoring, with a characteristic of longer trough-to-peak period and shorter peak-to-trough period (*Figure 2m*, *Figure 2—figure supplement 1c*). The similar phenotype of distorted bioluminescence rhythms was also observed in some mutant mice, such as $Tau$ mutant mice and $Afh$ mice (*Meng et al., 2008*; *Godinho et al., 2007*). Consistent with the ex vivo SCN explant data (*Figure 2i,h*), these results suggest that SIK3 is involved in circadian rhythm stability.

## PER2 expression levels are altered in *Sik3*-KD cells and *Sik3*-overexpressing cells

Previous reports indicate that phosphorylation-mediated modulation of clock protein stability is mediated by multiple protein kinases (*Reischl and Kramer, 2011*). To explore whether SIK3 modifies the phosphorylation status of the clock proteins, western blotting of cellular lysates expressing several types of clock proteins with different expression levels of SIK3 were performed. PER2 protein level was found to be largely affected by SIK3. $Sik3$-overexpression ($Sik3$-OX) in NIH3T3 cells resulted in a significant reduction in PER2 protein levels, but not PER1, compared with control cells (non-OX, *Figure 3a*, *Figure 3—figure supplement 1*). Cells expressing the constitutive-active SIK3 mutant T163E (*Katoh et al., 2006*) exhibited significantly reduced PER2 protein level in HEK293T17 cells (*Figure 3a*) and NIH3T3 cells (*Figure 3—figure supplement 1*), whereas in cells expressing the kinase-deficient SIK3 mutant K37M (*Katoh et al., 2006*) PER2 protein levels were comparable with those of control cells (*Figure 3a*, *Figure 3—figure supplement 1*). This suggests that the kinase activity of SIK3 is required for PER2 destabilization. Remarkably, it was observed that the relative level of the up-shifted PER2 band (relative to the total of the PER2 bands) was elevated in $Sik3$-OX cells compared with controls (*Figure 3b*, *Figure 3—figure supplement 2a*). Protein phosphatase (λPPase) treatment reduced these up-shifted PER2 bands, which confirmed that the shifted bands were phosphorylated forms of PER2 (*Figure 3c*). Further investigated was whether $Sik3$-OX induced PER2 instability or degradation. When protein synthesis was inhibited by treatment with cycloheximide (CHX), PER2 levels decreased rapidly in SIK3-OX cells compared with the control cells, indicating that SIK3 accelerates PER2 degradation (*Figure 3d*, *Figure 3—figure supplement 2b*). To further confirm SIK3-dependent degradation of PER2, four different shRNA constructs were used for $Sik3$ knockdown, and both PER2 abundance and its phosphorylation status were examined. All $Sik3$-KD cells expressing each $Sik3$ shRNA exhibited elevated levels of PER2 protein (*Figure 3e*, *Figure 3—figure supplement 3a,b*), and the relative levels of the up-shifted PER2 bands were significantly reduced (*Figure 3f*, *Figure 3—figure supplement 3c*). Reduced up-shifted band of PER2 by $Sik3$-KD was phosphorylation-dependent band, which was confirmed by λPPase treatment of these cells (*Figure 3g*). Degradation of PER2 protein levels in any of the $Sik3$-KD cells was significantly delayed, for as long as 6 hr after CHX treatment (*Figure 3h*). These data strongly suggest that SIK3 regulates PER2 abundance and stability by mediating its phosphorylation.

## Elevated PER2 protein levels are also observed in Sik3-/- fibroblasts and liver

To investigate whether alteration of endogenous PER2 protein levels is observed in the absence of SIK3, western blot analyses were performed on mouse embryonic fibroblasts (MEFs) isolated either from $Sik3^{-/-}$ or $Sik3^{+/+}$ mice (*Figure 4a*). Expression levels of PER2 were significantly elevated in

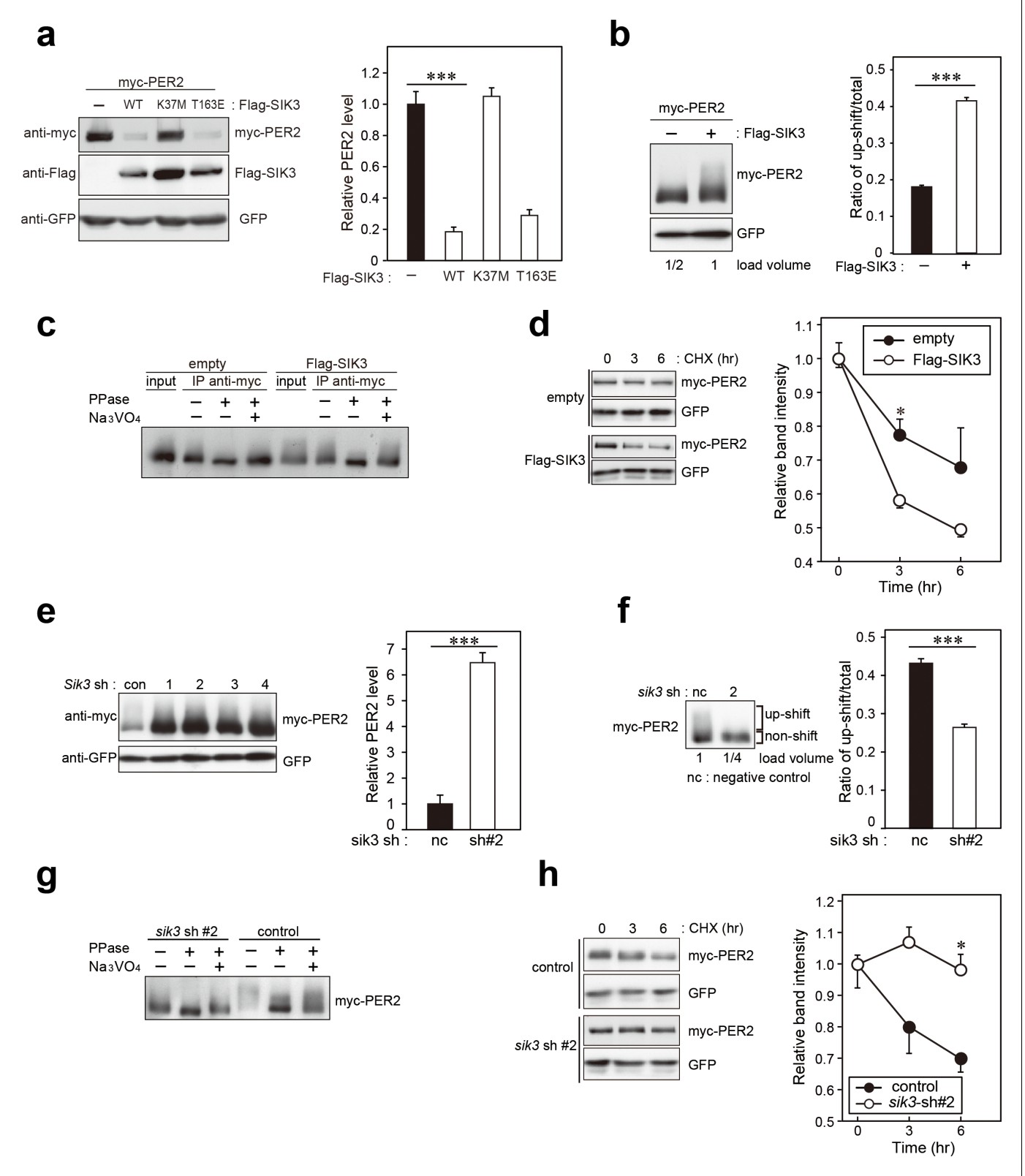

**Figure 3.** SIK3 alters expression levels and phosphorylation states of PER2 protein. (**a**) HEK293T17 cells were transfected with SIK3 and PER2 expressing constructs, and the effect on PER2 expression levels was examined by western blotting analysis. Overexpression of SIK3 (SIK3-OX) in HEK293T17 cells (WT) and constitutively active SIK3 cells (T163E), but not kinase-deficient SIK3 mutant cells (K37M), significantly reduced PER2 protein levels (n = 3, p<0.001 by Tukey's test). (**b**) Load volume of cell lysates in western blotting analysis was adjusted for comparison of phosphorylation rate. Adjustment

*Figure 3 continued on next page*

*Figure 3 continued*

of PER2 levels in SIK3-OX and non-OX controls revealed increased rates of upshifted PER2 protein (upshift/total) in NIH3T3 cells (n = 3, ***p<0.001 vs. controls by Student's *t*-test). (c) Myc-tagged PER2 co-expressed with SIK3 was purified from HEK293T17 cell lysate and incubated with λPPase for 30 min at 30°C. PER2 up-shift was decreased after λPPase treatment in SIK3-OX cells, which was attenuated by phosphatase inhibitor, $Na_3VO_4$. (d) PER2 degradation assay was performed in *Sik3*-OX cells. Cells were collected at 0, 3, and 6 hr after the addition of CHX. PER2 protein levels at the starting point (t = 0) were normalized to 1. All data used for quantification are shown in *Figure 3—figure supplement 2b* (n = 3, *p<0.05 by Student's *t*-test). *Sik3*-OX cells accelerated PER2 degradation. (e) NIH3T3 cells were transfected with each shRNAs of *Sik3* (SIK3-KD, #1–4) or negative control shRNA. All SIK3-KD significantly increased PER2 levels (n = 3, p<0.05 by Tukey's test). Another set of samples is also shown in *Figure 3—figure supplement 3a*. (f) *Sik3*-KD in NIH3T3 cells reduced PER2 up-shift. Load volume of cell lysates in western blotting analysis was adjusted for comparison of phosphorylation rate (n = 3 for control and shRNA#2, ***p<0.001 vs. controls by Student's *t*-test). (g) λPPase treatment with Myc-PER2 purified from cell lysates of control cells or *Sik3*-KD cells. PER2 up-shift was decreased after λPPase treatment in SIK3-OX cells, which was attenuated by phosphatase inhibitor, $Na_3VO_4$. (h) PER2 degradation assay in *Sik3*-KD cells was performed. PER2 protein levels at the starting point (t = 0) were normalized to 1 (n = 3, *p<0.05 by Student's *t*-test).

The online version of this article includes the following figure supplement(s) for figure 3:

**Figure supplement 1.** Constitutive active mutant, but not kinase-deficient SIK3, alters PER2 abundance in NIH3T3 cells.
**Figure supplement 2.** Overexpression of SIK3 promotes PER2 degradation.
**Figure supplement 3.** *Sik3* knockdown reduces degradation of PER2 and increases its levels.

*Sik3*[-/-] fibroblasts compared with *Sik3*[+/+] counterparts at all time points examined. To further examine the abnormal accumulation of PER2 in *Sik3*[-/-] mice in vivo, mouse liver was prepared from *Sik3*[-/-] and littermate *Sik3*[-/+] mice at ZT22, when PER2 protein levels and its phosphorylation levels are both high (*Figure 4b*, left). Western blot analyses demonstrated increased PER2 levels in *Sik3*[-/-] liver compared with the heterozygous mice (*Figure 4b*, right), which confirmed SIK3-dependent regulation of PER2 stability.

## Phosphorylation and degradation of PER2 by SIK3 through a CSNK1e-independent pathway

SIK3-catalyzed phosphorylation of PER2 was examined via an in vitro kinase assay. The up-shifted band of PER2 was significantly increased after incubation of PER2 with SIK3, and the total amount of PER2 was reduced in a manner dependent on the amount of SIK3 added (*Figure 5a*). PER2 protein

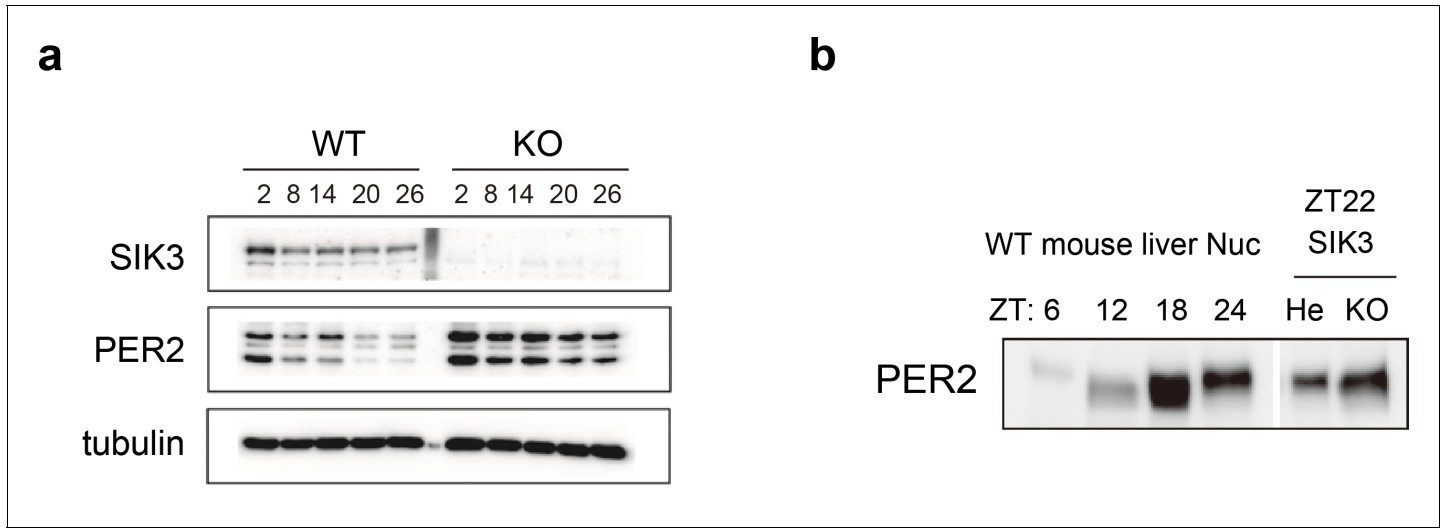

**Figure 4.** Elevated expression of PER2 in *Sik3*[-/-] fibroblasts and liver. (a) MEFs were established from *Sik3*[+/+] (WT) *Per2*[Luc/+] mice and *Sik3*[-/-] (KO); *Per2*[Luc/+] mice. The cellular rhythms were synchronized by dexamethasone (DEX, 100 nM) treatment to examine rhythmic expression by western blotting analysis. PER2 (bottom band) and PER2::LUC (upper bands) expression levels at all time points in knockout (KO) mouse embryonic fibroblast (MEFs) are higher than those in wild-type (WT) MEFs. (b) Western blotting performed on liver nuclear extracts, *Sik3*[+/-] (He); *Per2*[Luc/+] mice, and littermate *Sik3*[-/-] (KO); *Per2*[Luc/+] mice. PER2 expression in liver nucleus was higher in *Sik3*[-/-] mouse as compared to in *Sik3*[+/-] mouse ZT22, left). To confirm the protein mobility of endogenous PER2, rhythmic expression of PER2 in *Sik3*[+/+] (±) mice was also detected. Note that *Sik3*[+/+] (WT) mice do not have *Per2*[Luc] allele. Therefore, PER2 bands of *Sik3* He and KO are from a single *Per2* allele while those from WT are from both alleles.

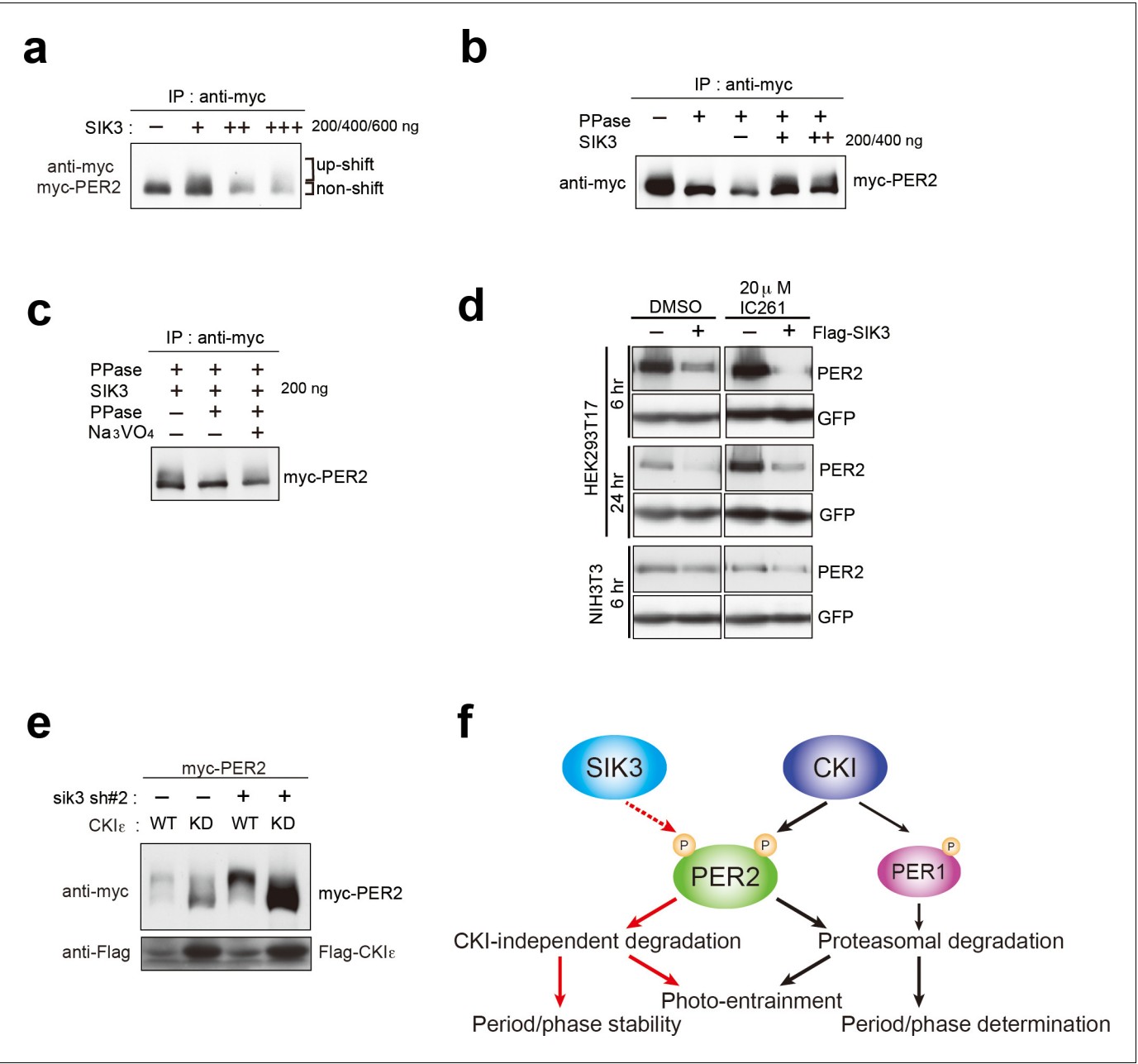

**Figure 5.** CSNK1-independent phosphorylation and destabilization of PER2 by SIK3. (**a**) In vitro kinase assays of Myc-PER2 purified from HEK293T17 cell lysate and recombinant SIK3 catalytic domain. Incubation of PER2 protein with SIK3 increased the up-shifted PER2 band. (**b**) Purified Myc-PER2 was treated with λPPase prior to a kinase assay with SIK3. No-phosphorylated PER2 was also phosphorylated by recombinant SIK3. (**c**) Up-shifted band shown in b was confirmed as phosphorylation-form of PER2 by 2nd λPPase assay. Purified Myc-PER2 from cell lysates was incubated with λPPase and then with SIK3 (200 ng). After washing SIK3 away, PER2 was subjected to secondary λPPase treatment. (**d**) NIH3T3 and HEK293T/17 cells expressing SIK3 were incubated with CSNK1 inhibitor (IC261) for indicated time periods. CSNK1 inhibitor did not affect SIK3-mediated PER2 degradation in both cell lines. (**e**) NIH3T3 cells were transfected with *Sik3*-KD constructs as well as *Csnk1e*-WT or kinase dead form of *Csnk1e* (*Csnk1e*-KD). Knockdown of *Sik3* did not affect CSNK1-mediated phosphorylation and degradation of PER2. (**f**) A schematic representation of a potential role for SIK3 compared with that of CSNK1. Although both SIK3 and CSNK1 mediate PER2 phosphorylation and degradation, their roles and degradation pathways appears to be different. Therefore, it may not be the case that SIK3 acts as a priming kinase for CSNK1, but the two kinases regulate PER2 independently. P represents phosphate.

level was decreased by degradation when PER2 was incubated with higher amounts of SIK3 (400 ng and 600 ng), even after purification of tagged PER2 from cell lysate. The effect of phosphatase treatment before SIK3 kinase assay was then examined. It is well known that CSNK1 requires a priming phosphorylation to bind to PER2 leading to further phosphorylation. In contrast, an in vitro phosphatase/kinase assay demonstrated that the down-shifting effect of phosphatase pre-treatment on the PER2 band was counteracted by the addition of SIK3 (*Figure 5b*). When PER2 was treated with λPPase after incubation with SIK3, the up-shifted PER2 band was significantly reduced, confirming that the up-shifted band was a form of PER2 that was phosphorylated via SIK3 signaling (*Figure 5c*). These data suggest that priming phosphorylation is not needed for SIK3-dependent phosphorylation and degradation of PER2, unlike CSNK1-mediated ones.

As mentioned above, CSNK1 is known to phosphorylate PER2 and induce PER2 degradation through a proteasomal pathway (*Akashi et al., 2002*; *Eide et al., 2005*; *Meng et al., 2008*). It has also been reported that CSNK1 requires an unidentified priming kinase prior to phosphorylate its substrates (*Gallego and Virshup, 2007*). To address the question of whether SIK3-induced phosphorylation of PER2 is independent of phosphorylation by CSNK1 (*Akashi et al., 2002*; *Eide et al., 2005*; *Meng et al., 2008*), NIH3T3 cells co-transfected with *Per2* and *Sik3* were incubated with the CSNK1 inhibitor IC261. IC261 did not affect PER2 degradation mediated by SIK3, suggesting that SIK3-dependent PER2 degradation is independent of CSNK1-mediated PER2 degradation (*Figure 5d*). Conversely, we performed KD experiments of *Sik3* and examined the effect on CSNK1-mediated phosphorylation and degradation of PER2. As shown in *Figure 3e*, KD of *Sik3* increased the basal level of PER2 proteins, but had no significant effect on the CSNK1-dependent phosphorylation states and degradation (*Figure 5e*). Collectively, these data indicated that PER2 phosphorylation mediated by two different kinases, CSNK1 and SIK3, are independent events.

## Discussion

In the current study, we found that SIK3 promotes phosphorylation of PER2 and regulates the abundance of the protein by accelerating its degradation. Interestingly, while both CSNK1d/e and SIK3-mediated phosphorylation of PER2 and promote its degradation, the underlying mechanisms appear to differ from one another (*Figure 5d and e*). However, two questions remain: how do they share or divide roles in terms of the regulation of PER2 stability and abundance, and why are two kinases that induce distinct degradation systems required for a common substrate? To address these questions, it is worth noting that $Sik3^{-/-}$ mice exhibit unique circadian phenotypes compared with those of *Tau*, a *Csnk1e* mutant, although both of these mutants also have features in common. While *Tau* mutant hamsters and mice demonstrate significant phase advance and shorter periods in physiological and behavioral rhythms (*Meng et al., 2008*; *Ralph and Menaker, 1988*), $Sik3^{-/-}$ mice, which are loss-of-function mutants, exhibit significant phase delay and longer circadian periods as shown in the current study (*Figure 1*). It has also been reported that *Csnk1d* KO liver exhibits significantly longer period in $Per2^{Luc}$ rhythms (*Etchegaray et al., 2009*). These data suggest that both SIK3 and CSNK1 contribute to precise circadian period determination, and that deficiency of either protein results in period shortening along with stabilization of PER2. By contrast, unique circadian phenotypes observed in $Sik3^{-/-}$ mice include the following: (i) fluctuating circadian periods and phases both in vivo (*Figure 1a–f*) and in vitro SCN cells (*Figure 2i and j*), (ii) impaired light-entrainment (*Figure 1d*), (iii) reduced amplitude in locomotor activity rhythm (*Figure 1g*), and (iv) desynchrony of rhythmicity among SCN cells (*Figure 2d,h*, *Figure 2—video 1*, *Figure 2—video 2*), none of which have been reported in *Csnk1* mutants, except that reduced amplitude in $Per2^{Luc}$ rhythms was observed in *Csnk1d* KO liver (*Etchegaray et al., 2009*). Thus, data from the current study suggest that while CSNK1 is involved in determination of precise circadian period length and phase, SIK3 is more important for stabilization of circadian periods and phase via intercellular coupling, as well as for photic entrainment (*Figure 5f*). In other words, while the two kinases may have an overlapping role(s) with regard to phosphorylating and destabilizing PER2, SIK3 and CSNK1 contribute differentially to circadian clock regulation.

It is also interesting to note that the above-described phenotypic differences between *Tau* mutants and $Sik3^{-/-}$ mice may, at least partly, be explained by differential roles of the two protein kinases outside of the SCN. For example, SIK3 is expressed in the eye (*Lizcano et al., 2004*) as well as in the SCN and, hence, disruption of this gene may affect the circadian light-input pathway

involving the retina, which is critical for photic-entrainment. This hypothesis, however, needs to be validated by further experiments such as a study investigating conditional KO mice to assess whether SIK3 in the eye plays a role in the entrainment. In addition, involvement of CSNK1 in the photic-entrainment cannot be excluded because of potential expression of *Csnk1* in the eye.

As shown in *Figure 1d and g*, in some cases, the actograms of *Sik3$^{-/-}$* mice suggest weakened rhythms. These data suggest the possibility that loss of SIK3-mediated signaling affects rhythm intensity, that is, less active at night and more active during the day. However, we found a wide variety of phenotypic differences in the individual *Sik3$^{-/-}$* mice. Because SIK3 is involved in the regulation of metabolism(s), as previously shown, and also because average blood glucose level in *Sik3$^{-/-}$* mice is significantly lower than that of controls, it may be a requirement for at least some of the KO mice to 'keep seeking food' even during a resting period, which may result in a change in rhythm strength. We believe that further studies, such as those investigating *Sik3* conditional KO mice or transgenic rescue mice, would enable us to distinguish the dual effects on behavioral rhythms (i.e. effects from the circadian clock and metabolism).

It is also noteworthy that light re-entrainment experiments shown in *Figure 1d*, some of the *Sik3$^{-/-}$* mice exhibited quite intriguing behaviors as mentioned above. It appears that, after transition from DD to LD, a part of the locomotor activities free-run even in the light phase (red dotted lines), but the rest of the activities were gradually entrained to LD cycles (*Figure 1d*, right), which is not observed in the controls (*Figure 1d*, left). There are a couple of possible interpretations: (i) 'relative coordination,' in which free-running rhythms are affected by a Zeitgeber (entrainment agent) or other rhythms; and (ii) incomplete re-entrainment and/or insufficient negative masking by light due to defect(s) in the light-input pathway.

It has previously been shown that constitutive expression of *Per2* causes arrhythmia in vitro and in vivo (*Chen et al., 2009*; *Yamamoto et al., 2005*). This may partly explain the aberrant rhythmicity in *Sik3$^{-/-}$* mice due to accumulation of PER2 protein. Desynchrony among SCN cells, which has been observed in circadian rhythm mutants (*Maywood et al., 2006*), may lead to insufficient light-entrainment or inconstant behavioral rhythms. Finally, the *Sik3* gene is highly conserved in evolution, from nematodes to mammals (*Okamoto et al., 2004*), and *Sik3* deficiency results in severe phenotypes such as high lethality after birth, and metabolic and skeletal abnormalities (*Sasagawa et al., 2012*; *Uebi et al., 2012*). These previous studies suggest that SIK3 plays an essential role in various tissues and organs. Considering that animal behavior is influenced by physiological states, we cannot exclude the possibility that behavioral and physiological abnormalities in circadian rhythms in *Sik3$^{-/-}$* mice, such as those observed in the current study, are due to defects not only in the SCN, but in other parts of the brain or in other organs or tissues. Conditional *Sik3$^{-/-}$* mouse may be a suitable tool for addressing these questions in the future. In addition, the present study demonstrated that SIK3-mediated phosphorylation regulates protein stability of PER2; however, this does not exclude the possibility that other clock/clock-related protein(s) is also a target of SIK3.

It has been reported that SIK1, another isoform of the SIK subfamily, is involved in photic entrainment of the master circadian clock (*Jagannath et al., 2013*), although its involvement in the central circadian clock remains uncertain. Notably, while the present study demonstrated that SIK3 regulates photic-entrainment, light responses in behavior observed in *Sik1*-KD mice (*Jagannath et al., 2013*) and *Sik3$^{-/-}$* mice are contrary to one another: rapid *vs.* delayed re-entrainment, respectively. In addition, CREB-regulated transcription coactivator 1 (CRTC1) was proposed as a target of SIK1, which contributes to light-induced phase reset (*Jagannath et al., 2013*). Although further studies are necessary to elucidate the involvement of the two SIK isoforms in light-resetting mechanisms, the current study suggests that SIK3 contributes to the light-input pathway in addition to the circadian clock machinery in a manner that differs from SIK1, by regulating different substrates and signaling pathways.

Strikingly, Funato *et al.* recently reported that, by large-scale forward-genetics screen in mice to seek dominant sleep abnormalities, a splicing mutation of *Sik3* (referred to as a *Sleepy* mutant) resulted in significant decrease in wake time caused by extended non-REM (NREM) sleep (*Funato et al., 2016*). In the *Sleepy* mutants in which a point mutation resulting in a lack of a protein kinase A (PKA) recognition site of the SIK3 protein, the authors also described that locomotor activity rhythms in the mutants were normal. Although the results were in contrast to our present data, the discrepancy could be explained by a difference in introduced mutations, that is, a partial deletion in the *Sleepy* mutants versus null mutation in our *Sik3* KO mice.

More recently, a study in *Drosophila* suggested effects of SIK3 homolog on coupling of different oscillator cells and with alternative period-lengthening and shortening in different cell types (*Fujii et al., 2017*). Interestingly, the present study in mice also observed abnormality in extracellular coupling among clock cells in *Sik3*-knockout mice (*Figure 2*). Fujii *et al.* described clearly that the amplitude of dPER protein rhythms in clock neurons (DN1p) was damped in *Sik3*-knockdown flies under constant darkness, while they did not propose a role for SIK3 in the phosphorylation and degradation of dPER. Instead, they proposed HDAC4 as a target of SIK3 since HDAC4 is one of well-established targets of SIK3 in fly. Thus, it is possible that SIK3 regulates the mammalian circadian clock by modulating activity of HDAC4 besides PER2 or that the effect of SIK3 on PER2 degradation is mediated by some kinases or degradation proteins, which are transcriptionally regulated by HDAC4.

## Materials and methods

### Mice

All animals were cared for in accordance with Law No. 105 and Notification No. 6 of the Japanese Government, and experimental protocols involving the mice were approved by the relevant Ethics Committees at Kindai University and Nagoya University. Mice were maintained at 23 ± 1°C under LD 12:12 hr or constant darkness (DD conditions). The derivation of the $Sik3^{-/-}$ mice (RRID: MGI: 5317926) has been reported previously (*Uebi et al., 2012*). Briefly, the PGK-neo cassette was inserted in place of exon 1 of *Sik3*. The successful targeting of embryonic stem cells was confirmed by Southern blot analysis, and the cells were injected into C57BL/6N blastocysts. To obtain *Sik3*-deficient embryos, *Sik3* heterozygous male and female mice were mated and WT littermates were used as controls. *Per2::LUC* knockin mice (RRID: IMSR_JAX:006852) were originally generated by Joseph Takahashi (*Yoo et al., 2004*). All mice used in the experiments were mated with C57BL/6J mice (SLC, Japan). Genotyping was performed using genomic DNA from a tail tip of each mouse via a standard protocol as previously described (*Sasagawa et al., 2012*; *Yoo et al., 2004*). Analyses of metabolic rhythms were analyzed as previously described (*Uebi et al., 2012*). Briefly, voluntarily O2 consumption ($VO_2$) of each group was monitored for two days using the Oxymax system (Columbus Instruments, Columbus, Ohio), and the average was calculated every hour (n = 6). Food consumption and rectal temperature rhythms were measured manually and averaged every 6 hr (n = 12 each).

### Behavioral analyses

Mice were individually housed in translucent polypropylene cages under 12:12 hr LD (200 lux) and DD conditions, and locomotor activity was assessed by an area sensor (infrared sensor, Omron, Japan). Activity was continuously monitored and analyzed using ClockLab software (Actimetrics, Wilmette, IL, RRID: SCR_014309).

### In situ hybridization

Riboprobe was labeled with $^{35}$S-UTP (Amersham/GE Healthcare, Pittsburgh, PA) by in vitro transcription using either T7 or SP6 polymerase (Promega, Madison, WI). Frozen mouse brain sections (40 µm thick) were hybridized with riboprobe overnight, and exposed to Kodak film (BioMax MR, Eastman Kodak, Rochester, NY). The *Sik3* cDNA (550 bp) was amplified via PCR and subcloned into a pGEM-T Easy Vector (Promega). The plasmids were linearized with NcoI to synthesize riboprobe.

### Bioluminescence measurement, imaging, and quantitative analysis

For bioluminescence recording, NIH3T3 cells (3T3-3-4, Riken Cell Bank, Japan, RRID: CVCL_1926) transfected with *Bmal1-luc* in a 35 mm cell culture dish were used. NIH3T3 cell line was authorized as follows: identification of animal species was done by PCR and isozyme analysis and mouse strain was identified by SSLP analysis.

For SCN imaging, adult $Sik3^{-/-}$; $Per2^{Luc/Luc}$ and $Sik3^{+/+}$; $Per2^{Luc/Luc}$ mice were used. Cellular rhythms were synchronized with 100 nM dexamethasone (DEX). Two hours after the DEX treatments, the culture medium was replaced with a recording medium, DMEM culture medium including 10 mM HEPES (pH 7.0), 10% fetal calf serum (FCS) and 0.1 mM luciferin, and circadian bioluminescence rhythm was monitored in Kronos (ATTO, Japan). For single-cell time-lapse imaging, LV 200

(Olympus, Japan) was used for SCN slices. SCN slices from adult $Sik3^{-/-}$; $Per2^{Luc/Luc}$ and $Sik3^{+/+}$; $Per2^{Luc/Luc}$ mice were prepared using a Microslicer (Dosaka Japan, 300 µm thick) and incubated on MilliCell membrane (Merck Millipore, Billerica, Massachusetts) in a 35-mm petri dish with culture medium (DMEM-F12 supplemented with B27, Thermo Fisher Scientific, Waltham, MA). Exposure time for a single image was 55 min, and time-lapse images were captured for five consecutive days. Movies were produced using MetaMorph software (Molecular Devices, Sunnyvale, CA).

## Analysis of period and acrophase in SCN slices

For each pixel of the bioluminescence movie, the time series was extracted. Each data-set was detrended and analyzed via the chi-squared periodogram (*Sokolove and Bushell, 1978*). Only periods with a significance level of less than 10% were plotted (*Figure 2*). The acrophase was subsequently computed via Cosinor's method (*Halberg et al., 1967*), which extracts a peak phase of the bioluminescence signal of each pixel. Each pixel's phase relative to that of the averaged signal over the whole SCN slice was plotted (*Figure 2*).

## Cell culture and plasmids for transfection

NIH3T3 (3T3-3-4, Riken Cell Bank, RRID: CVCL_1926) and HEK293 (HEK293T/17, ATCC, Japan, RRID: CVCL_L993) were authorized by Riken Cell Bank and ATCC respectively as follows. NIH3T3 was authorized at Riken Cell Bank by PCN and isozyme analysis for identification of animal species, and by SSLP analysis for mouse strain. HEK293T/17 was authorized at ATCC by STR (Short Tandem Repeat polymorphism analysis). These cell lines were cultured and passaged under 5% $CO_2$ in DMEM (Sigma, St. Louis, MO) containing 1.8 mg/ml $NaHCO_3$, 4.5 mg/ml glucose, 100 U/ml penicillin, 100 µg/ml streptomycin, and 10% FCS (Equitech BioInc, Kerrville, TX). Mycoplasma contamination was examined for both cell lines.

NIH3T3 and HEK293 cells were transiently transfected using Lipofectamine Plus reagent (Invitrogen, Carlsbad, CA) and Lipofectamine 2000 reagent (Invitrogen), respectively, in accordance with the manufacturer's instructions. The pEGFP-C1 plasmid (Clontech, Mountain View, CA) was used as a transfection control. A vector encoding 6xMyc epitope-tagged PER2 (termed myc-PER2 in the figures) was kindly provided by Dr. Louis J. Ptacek (University of California, San Francisco). For Flag-SIK3, an oligonucleotide encoding the FLAG epitope sequence was fused to the 5′-end of full-length mouse *Sik3* (m*Sik3*) cDNA cloned into pcDNA 3.1. Mutations (Lys to Met at K37 or Thr to Glu at T163) were introduced into m*Sik3* using site-directed mutagenesis. Flag-Csnk1e (CKIε) and Flag-Csnk1d−KN (Kinase Negative) have been described previously (*Doi et al., 2004*). Several shRNA expression vectors targeting m*Sik3* gene and a control vector were purchased from QIAGEN (Germany). The following sequences were used: m*Sik3* sh#1 (5′- CTGCA GGCAC AAGTG GATGA A-3′), m*Sik3* sh#2 (5′- AGCAG CAACC CGAGA ACTGT T-3′), m*Sik3* sh#3 (5′- CCCAA CTTTG ACAGG TTAAT A-3′), m*Sik3* sh#4 (5′- TGCCA CCACG TTCAG TAGAA A-3′), and control sh (5′- GGAAT CTCAT TCGAT GCATA C-3′). The target sequences were inserted into a pGeneClip neomycin vector (Promega).

## Immunoblotting

Proteins separated by SDS-PAGE were transferred to polyvinylidene difluoride membrane (Millipore). The blots were blocked in a blocking solution (1% [w/v] skim milk in PBS-T [50 mM Tris-HCl, 140 mM NaCl, 1 mM $MgCl_2$; pH 7.4]) for 1 hr at 37°C then incubated 1 hat 37°C or overnight at 4°C with a primary antibody in the blocking solution. Signals were visualized via an enhanced chemiluminescence detection system (PerkinElmer Life Science, Waltham, MA). The blot membrane was subjected to densitometric scanning, and the band intensities were quantified using Image Quant software (GE Healthcare). The primary antibodies used were anti-β-actin (Sigma), anti-Flag (Sigma), anti-myc (Santa Cruz Biotechnology, Dallas, TX), anti-GFP (Santa Cruz Biotechnology), and anti-PER2 (Millipore for *Figure 4a* and Alpha Diagnostic Inc. for *Figure 4b*). Primary antibodies were detected by horseradish peroxidase-conjugated anti-rabbit or anti-mouse IgG (Kirkegaard and Perry Laboratories).

## Immunoprecipitation

Transfected cells were lysed for 30 min in ice-chilled immunoprecipitation (IP) buffer (20 mM HEPES-NaOH, 137 mM NaCl, 2 mM EDTA, 10% [v/v] glycerol, 1% [v/v] Triton X-100, 1 mM DTT, 4 μg/ml aprotinin, 4 μg/ml leupeptin, 50 mM NaF, 1 mM $Na_3VO_4$, 1 mM phenylmethylsulfonyl fluoride; pH 7.8). The lysate was incubated with the precipitating antibody for 2 hr at room temperature or overnight at 4°C, followed by incubation with 20 μl of Protein G-Sepharose beads (GE Healthcare) for 2 hr at 4°C. Beads were washed three times with IP buffer and subjected to immunoblotting.

## Degradation assay

NIH3T3 cells were transfected with myc-PER2 and Flag-SIK3 expression vectors or *Sik3*-KD vectors, and cultured for 48 hr. The transfected cells were then treated with cycloheximide (Nakalai tesque, Japan     ; final concentration, 100 μg/ml) for the time periods specified in the figures *Figure 3*, *Figure 3—figure supplement 2*), and then harvested, followed by immunoblotting.

## Phosphatase assay

NIH3T3 cells were transfected with myc-PER2 and Flag-SIK3 expression vectors or *Sik3*-KD vectors, and cultured for 48 hr. Immunoprecipitated myc-PER2 were incubated with 400 IU λPPase (Sigma) in Lambda Protein Phosphatase Buffer (Sigma) containing 2 mM $MnCl_2$ for 30 min at 30°C in the presence of Protein G Sepharose. When samples were sequentially subjected to kinase assay, λPPase had been removed by washing PER2 bind to Protein G Sepharose in kinase assay buffer three times. $Na_3VO_4$ was used for a phosphatase inhibitor.

## In vitro kinase assay

NIH3T3 cells were transfected with myc-PER2 vector and cultured for 48 hr. Immunoprecipitated myc-PER2 were incubated with GST-tagged recombinant human SIK3 (aa.1–307) (SignalChem, Canada) in kinase assay buffer (50 mM Tris-HCl, 10 mM $MgCl_2$, 100 mM NaCl, 1 mM DTT, 10% glycerol, 1 mM ATP; pH 7.4) at 30°C for 30 min or 75 min. When samples were sequentially subjected to phosphatase assay, SIK3 had been removed by washing PER2 bind to Protein G Sepharose in phosphatase assay buffer three times.

## Statistics

All data are expressed as mean ± SEM or SD. Statistical significance was evaluated via Student's *t*-test, and was set at the $p < 0.05$ level. One-way ANOVA followed by Dunnett's test or Turkey's test was used for multiple comparisons. For *Figures 1f*, *2i and j*, tests for equality of variance (F-test) were performed, and statistical significance was set at the $p < 0.05$ level. We did not use statistical analysis to estimate sample size and follow the protocols because our experimental methods have been commonly used.

Number of animals used for experiments are all independent repeats. In KD experiments in *Figure 2* and Suppl. *Figure 2*, we performed two independent experiments for PER2 abundance and phosphorylation, each of which include triplicates. Degradation assay using shRNAs were performed one time each (triplicates), using three different shRNAs (ShRNAs #1, 3, and 4). As for overexpression (*Sik3*-OX) experiments, quantification of PER2 protein was independently done twice using triplicates, and two different cell lines (NIH3T3 and HEK 293) were used in the same series of experiments for confirmation. Phosphorylation of PER2 in SIK3-OX 3T3 cells were examined by two independent experiments, whereas PER2 degradation was done once (n = 3). Circadian rhythms in KD cells were analyzed twice, each of which shRNAs, #1, 3, and 4 were used.

## Acknowledgements

The authors are grateful to Drs. Hiroshi Takemori and Minako Koura for providing *Sik3* KO mice and their data on metabolic rhythms (*Figure 1a,b*), Dr. Joseph Takahashi for providing *Per2^Luc* KI mice, and Drs. Daisuke Ono, Sato Honma, and Ken-ichi Honma for supporting bioluminescence imaging. They also thank Dr. Mamoru Nagano, Yuka Sugahara, Mika Machida, and Nana Itou for technical assistance.

## Additional information

### Funding

| Funder | Grant reference number | Author |
|---|---|---|
| Japan Society for the Promotion of Science | Grant-in-Aid for Scientific Research 25293053 | Naoto Hayasaka |
| Japan Science and Technology Agency | PRESTO | Naoto Hayasaka |
| Japan Society for the Promotion of Science | Grant-in-Aid for Scientific Research 24227001 | Yoshitaka Fukada |
| Japan Society for the Promotion of Science | Grant-in-Aid for Scientific Research 17H06096 | Yoshitaka Fukada |

The funders had no role in study design, data collection and interpretation, or the decision to submit the work for publication.

### Author contributions

Naoto Hayasaka, Conceptualization, Data curation, Formal analysis, Funding acquisition, Investigation, Writing—original draft, Project administration, Writing—review and editing; Arisa Hirano, Investigation, Writing—original draft, Writing—review and editing; Yuka Miyoshi, Isao T Tokuda, Data curation, Investigation; Hikari Yoshitane, Data curation, Investigation, Writing—review and editing; Junichiro Matsuda, Supervision, Investigation, Project administration; Yoshitaka Fukada, Data curation, Formal analysis, Funding acquisition, Investigation, Writing—original draft, Writing—review and editing

### Author ORCIDs

Naoto Hayasaka  http://orcid.org/0000-0003-2844-524X
Arisa Hirano  http://orcid.org/0000-0001-9732-6100
Isao T Tokuda  http://orcid.org/0000-0001-6212-0022
Hikari Yoshitane  http://orcid.org/0000-0001-6319-3354

### Ethics

Animal experimentation: This study was performed in strict accordance with the recommendations in the Guide for the Care and Use of Laboratory Animals of the Japan Society for Promotion of Sciences. All of the animals were handled according to approved institutional animal care and use committees of Kindai University (KAME- 19-051) and Nagoya University (17239).

### Decision letter and Author response

Decision letter https://doi.org/10.7554/eLife.24779.sa1
Author response https://doi.org/10.7554/eLife.24779.sa2

## Additional files

### Supplementary files

- Transparent reporting form

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
