## [Decision Letter]

Thank you for submitting your article "Salt-inducible kinase 3 regulates the mammalian circadian clock by destabilizing PER2 protein" for consideration by *eLife*. Your article has been favorably evaluated by a Senior Editor and three reviewers, one of whom is a member of our Board of Reviewing Editors.

The reviewers have discussed the reviews with one another and have raised some significant concerns that we feel must be addressed.

Summary:

The study by Hayasaka et al. reports a role for the salt-inducible kinase 3 (SIK3) in the circadian clock of mice. SIK3 KO mice display slightly longer periods in locomotor activity rhythms and less efficient entrainment to light dark cycles. The circadian periods of SCN neurons within a cultured slice, however, are not significantly different from wild-type, yet the phase coherence of individual neurons seems to be affected. In contrast to behavior, circadian periods of NIH3T3 fibroblasts with RNAi-mediated knockdown of SIK3 are shorter than in control cells. The authors then test the effect of SIK3 levels on the abundance and stability of PER2 and find that overexpressing SIK3 (but not a kinase-dead version) decreases ectopically expressed PER2 levels by destabilization, and knockdown of SIK3 increases ectopically expressed PER2 levels. In addition, the authors show that SIK3 overexpression leads to an increase in overexpressed PER2 phosphorylation, and the resulting destabilization is independent from CK1 phosphorylation and proteasomal degradation.

Essential revisions:

The study is potentially interesting, but all three of the reviewers were concerned about the central hypothesis that PER2 is the relevant substrate for the circadian effects of this kinase. The effects of RNAi knock-down or overexpression on PER2 levels in tissue culture are very strong, and yet the period lengthening in the SIK3 knock-out mouse are very small (and go in the opposite direction from those seen with RNAi in NIH 3T3 cells) and therefore inconsistent with the strong effects seen in tissue culture manipulations.

1) Why did the authors investigate PER2 as a target for Sik3? Did they investigate any other circadian targets and fail to find effects on those? Could the direct target be something besides PER2 that produces a downstream effect on PER2?

2) There was a consensus that some additional findings are needed to strengthen the conclusion that PER2 is a relevant target, and that these findings should employ the SIK3^-/-^ cells rather than overexpression or RNAi studies. These studies could include immunoblot analysis of PER2 levels in the SIK3^-/-^ knock-out mice or in primary fibroblasts from SIK3^-/-^ mice to show that these levels are significantly higher than in wild type mice, with blunted oscillations and effects on the PER2 phosphorylation profile. The plots of overall per2-luciferase levels (rather than normalized levels) could also be informative about differential effects on PER2. Primary fibroblasts derived from the SIK3^-/-^ per2-luciferase mice (and SIK3^+/+^ per2 luciferase control fibroblast) could be assessed for short or long period oscillations, rather than the NIH 3T3 cells with RNAi kd, and an interaction of PER2 with SIK3 in normal circadian cells would also be indicative of PER2 relevance (although an absence of such an interaction would not be informative, as many kinases do not interact stably with their substrates). Identification of a SIK3 phosphorylation site in PER2 and demonstration of a destabilizing effect of phosphorylation at these sites would be a seminal finding, although we do not insist that this finding be produced for this manuscript.

3) In any event, additional findings in circadian cells that do not involve overexpression or RNAi are needed, and if the additional findings do not show strong effects on PER2 the authors must discuss PER2 as a target of Sik3 in a speculative manner.

[Editors' note: further revisions were requested prior to acceptance, as described below.]

Thank you for resubmitting your work entitled "Salt-inducible kinase 3 regulates the mammalian circadian clock by destabilizing PER2 protein" for further consideration at *eLife*. Your revised article has been favorably evaluated by a Senior Editor and a Reviewing Editor.

The manuscript has been improved but there are some remaining issues that need to be addressed before acceptance, as outlined below:

Hayasaka et al. have made a thorough effort to revise the manuscript according to the suggestions of the previous reviewers, and the paper is much improved. However, there are still a few points that need to be addressed. Most importantly, a paper describing the role of the SIK3 kinase in the *Drosophila* clock has been published (Fujii et al., Proc Natl Acad Sci U S A. (2017) 114(32):E6669-E6677) since the initial submission of this manuscript. This manuscript is not cited or discussed in the revised version of this manuscript under review, but it needs to be, particularly since some of the findings (e.g., effects of the kinase on coupling of different oscillator cells and with alternative period-lengthening and shortening in different cell types) are also found in this revised manuscript.

In contrast with this revised manuscript, the *Drosophila* manuscript does not propose a role for SIK3 in the phosphorylation and degradation of PER. Instead, HDAC4 is proposed as a target. The *Drosophila* paper did not investigate thoroughly an effect on PER levels and phosphorylation, so it may have missed them (feel free to suggest this), or alternatively the *Drosophila* clock may work differently from the mammalian clock with regard to this kinase. Nevertheless, I think the authors of this revised submitted manuscript need to be cautious about claiming direct regulation of PER2 phosphorylation and stability by SIK3 kinase activity. Clearly, PER2 is less stable and more phosphorylated with catalytically active SIK3 overexpression and more stable and less phosphorylated with SIK3 knock-down or expression of the inactive kinase, but their attempts to identify a SIK3 target site in PER2 or a protein degradation mechanism that mediates this effect were not successful. The "in vitro" phosphorylation of immunoprecipitated PER2 by SIK3 kinase in Figure 5A by their own admission involves other factors besides PER2 and SIK3 (e.g., proteolysis components), so it might also involve an intermediate kinase that responds to SIK3 addition to phosphorylate PER2. I think it is fair to say that the kinase activity of SIK3 leads to the phosphorylation and degradation of PER2, but not necessarily directly. It could be mediated by other kinases or other clock components, which do not appear to have been assessed extensively as targets for Sik3 (just PER1). If HDAC4 is a target for SIK3, its activity could modulate the expression of many other kinases and degradation proteins that contribute to the effect on PER2. I recommend modifying the last sentence of the Abstract to "Collectively, the results indicate that SIK3 plays key roles in circadian rhythms by facilitating phosphorylation-dependent destabilization of PER2, either directly or indirectly." Also, offer the alternative of an indirect regulation of PER2 phosphorylation and degradation status by SIK3 in the Results presentation of the Figure 5 data and in the Discussion, which now suggests only that PER2 is directly phosphorylated by SIK3.

---

## [Author Response]

Essential revisions:The study is potentially interesting, but all three of the reviewers were concerned about the central hypothesis that PER2 is the relevant substrate for the circadian effects of this kinase. The effects of RNAi knock-down or overexpression on PER2 levels in tissue culture are very strong, and yet the period lengthening in the SIK3 knock-out mouse are very small (and go in the opposite direction from those seen with RNAi in NIH 3T3 cells) and therefore inconsistent with the strong effects seen in tissue culture manipulations.1) Why did the authors investigate PER2 as a target for Sik3? Did they investigate any other circadian targets and fail to find effects on those? Could the direct target be something besides PER2 that produces a downstream effect on PER2?

Thank you for these insightful comments and questions. We investigated other proteins, including some clock proteins, and initially found that PER2 protein levels were largely affected by SIK3. Therefore, we commenced the PER2 study and found that, in in vitro kinase assays, incubation of PER2 purified from cells and recombinant SIK3 enhanced PER2 phosphorylation, indicating that PER2 is a direct substrate of SIK3-catalyzed phosphorylation. However, we do not exclude the possibility that other clock/clock-related protein(s) is/are also a target of SIK3. In the revised manuscript, we add the following statement to the Discussion:

“In addition, the present study showed that PER2 protein is a substrate of SIK3-mediated phosphorylation; however, this does not exclude the possibility that other clock/clock-related protein(s) is also a target of SIK3.”

2) There was a consensus that some additional findings are needed to strengthen the conclusion that PER2 is a relevant target, and that these findings should employ the SIK3^-/-^ cells rather than overexpression or RNAi studies. These studies could include immunoblot analysis of PER2 levels in the SIK3^-/-^ knock-out mice or in primary fibroblasts from SIK3^-/-^ mice to show that these levels are significantly higher than in wild type mice, with blunted oscillations and effects on the PER2 phosphorylation profile. The plots of overall per2-luciferase levels (rather than normalized levels) could also be informative about differential effects on PER2. Primary fibroblasts derived from the SIK3^-/-^ per2-luciferase mice (and SIK3^+/+^ per2 luciferase control fibroblast) could be assessed for short or long period oscillations, rather than the NIH 3T3 cells with RNAi kd, and an interaction of PER2 with SIK3 in normal circadian cells would also be indicative of PER2 relevance (although an absence of such an interaction would not be informative, as many kinases do not interact stably with their substrates). Identification of a SIK3 phosphorylation site in PER2 and demonstration of a destabilizing effect of phosphorylation at these sites would be a seminal finding, although we do not insist that this finding be produced for this manuscript.

Thank you for these comments and constructive suggestions. As mentioned in the previous response, we established mouse embryonic fibroblast (MEF) cell lines from SIK3^-/-^; PER2::LUC^+/-^ and SIK3^+/+^; PER2::LUC^+/-^ embryos. In western blot analysis, we found that endogenous PER2 protein levels were markedly elevated in SIK3^-/-^ MEFs at all of the time-points studied. This result strengthened our conclusion that SIK3 promotes the degradation of PER2 protein. This finding is included in revised manuscript as the new Figure 4A.

In addition, we attempted to detect/determine PER2 protein levels in SIK3-KO mouse liver. As mentioned, many SIK3-KO mice died on the day of birth, and only one SIK3^-/-^; PER2::LUC^+/-^ mouse was available for this experiment. Unfortunately, no SIK3^+/+^; PER2::LUC^+/-^ littermates were produced; therefore, we used a Sik3^+/-^; PER2::LUC^+/-^ littermate as a control in this experiment. Of note, we observed a dramatic increase in PER2 protein level in SIK3-KO mouse liver compared with control SIK3-heterozygous mice (Figure 4B in the revised manuscript, compare lane 5 and 6).

In response to the latter comment, we tried to identify PER2 phosphorylation site(s) responsible for control of its degradation rate. For this purpose, we prepared four single amino-acid (SA) mutants of the PER2 protein, in which each of four potential phosphorylation sites (all Serine) by SIK3 was mutated to Alanine (i.e., S624A, S659A, S971A, and S981A). These phosphorylation sites were identified by MS analysis in cell culture (Vanselow et al., 2007). Among all sites, S624 was identified by omics studies most frequently (PhosphoSitePlus, https://www.phosphosite.org/homeAction.action) and S659 is a well-known site regulating PER2 protein stability (Toh et al., 2001). We also considered the kinase consensus sequence of SIK3 (LXR[T/S]XpSXXX), especially regarding S971 and S981. Because the responsible kinases for all four sites are undetermined, we focused on these sites as candidates for SIK3-mediated phosphorylation.

We compared the steady state protein levels of the mutants with wild-type (WT)-PER2 in the presence of co-expressed SIK3 in HEK293T17 cells. However, no significant stabilization of PER2 protein was detected in any of the single SA mutations of PER2 (Author response image 1). Combination of these SA mutations and other sites should be examined in the future (because MS analysis of SIK3-mediated PER2 phosphorylation sites appear to be difficult due to their unstable nature).

**Author response image 1. respfig1:** No significant effects of SA mutations at phosphorylation sites of PER2 on its degradation induced by SIK3. HEK293T17 cells were transfected with SIK3 (or empty vector [–]) and mutant PER2 expression vectors with green-fluorescent protein (GFP) expression vector. The PER2 and GFP protein levels were analyzed using western botting. WT, Wild type.

3) In any event, additional findings in circadian cells that do not involve overexpression or RNAi are needed, and if the additional findings do not show strong effects on PER2 the authors must discuss PER2 as a target of Sik3 in a speculative manner.

Thank you, these are valid points. As mentioned above, we observed markedly elevated expression levels of PER2 in SIK3^-/-^ cells (new Figure 4A in revised manuscript) and SIK3^-/-^ liver (new Figure 4B in revised manuscript), which strongly suggest that PER2 is the most promising SIK3 substrate candidate.

[Editors' note: further revisions were requested prior to acceptance, as described below.]

[…] I recommend modifying the last sentence of the Abstract to "Collectively, the results indicate that SIK3 plays key roles in circadian rhythms by facilitating phosphorylation-dependent destabilization of PER2, either directly or indirectly." Also, offer the alternative of an indirect regulation of PER2 phosphorylation and degradation status by SIK3 in the Results presentation of the Figure 5 data and in the Discussion, which now suggests only that PER2 is directly phosphorylated by SIK3.

Thank you for these insightful comments. While we have tried to indicate the direct effect of SIK3 on PER2, we agree that our data is not strong enough to conclude that. According to your suggestion, we have modified our statements in the Abstract for the role of SIK3 as follow according to the reviewers’ suggestion:

“Collectively, SIK3 plays key roles in circadian rhythms by facilitating phosphorylation-dependent destabilization of PER2, either directly or indirectly.”

In addition, we cited the paper that reviewers had suggested (Fujii et al., Proc Natl Acad Sci USA. (2017) 114(32): E6669-E6677) and added the following statement to the Discussion:

“More recently, a study in *Drosophila* suggested effects of SIK3 homolog on coupling of different oscillator cells and with alternative period-lengthening and shortening in different cell types (Fujii et al., 2017). […] Thus, it is possible that SIK3 regulates the mammalian circadian clock by modulating activity of HDAC4 besides PER2 or that the effect of SIK3 on PER2 degradation is mediated by some kinases or degradation proteins, which are transcriptionally regulated by HDAC4.”